# The 8-Week Efficacy of Frequency Rhythmic Electrical Modulated System (FREMS) as an Add-on Therapy in the Treatment of Symptomatic Diabetic Peripheral Polyneuropathy

**DOI:** 10.3390/ijerph20010111

**Published:** 2022-12-22

**Authors:** Daria Gorczyca-Siudak, Piotr Dziemidok

**Affiliations:** Faculty of Human Sciences, University of Economics and Innovation, St. 4 Projektowa, 20-209 Lublin, Poland

**Keywords:** diabetes, peripheral neuropathy, electrical stimulation

## Abstract

Background: Frequency Rhythmic Electrical Modulated System (FREMS) is a method of transcutaneous treatment based on frequency-modulated electromagnetic neural stimulation. Its efficacy in neuropathic pain in diabetes mellitus still lacks enough research. Methods: A randomized, single-blind, sham-controlled trial in individuals with symmetric distal polyneuropathy (SDPN) as an add-on therapy compared to standard therapy with alpha-lipoic acid. Participants were randomized to FREMS and standard of care (SOC) versus SOC only. The primary outcome was a change from baseline in perceived pain assessed by visual analogue scale (VAS) after 5 days of treatment and after 8 weeks of follow-up between treatment groups. Results: After 5 days of treatment, patients in both groups felt significant reduction in pain as measured by VAS, although only FREMS treatment lasted for 8 weeks and induced a significant improvement in quality of life measured by EuroQol 5-Dimension 5-Level (EQ-5D-5L) and Clinical Global Impression of Change (CGI-C) questionnaires. There were non-significant differences observed in the instrument pain assessment. No relevant side effects were recorded during the study. Conclusions: FREMS as an addition to alpha-lipoic acid therapy occurred to be a beneficial method of treatment in individuals with SDPN and was associated with improvements in pain severity, quality of life and clinical global improvement.

## 1. Introduction

Diabetes mellitus (DM) is a chronic disease and a growing public health problem. It can lead to severe and progressive macro- and micro-vascular complications and results in increased mortality and decreased quality of life. One of diabetes chronic microvascular complications is a symmetric symptomatic diabetic peripheral neuropathy (DSPN). It is an important diabetic complication with a lifetime prevalence of more than 50% among people with diabetes. Among DSPN symptoms, neuropathic pain (painful DSPN) affects up to 30% of all individuals [1,2]. Neuropathic pain is often chronic, severe and difficult to treat and manage. DSPN is the most common form of neuropathy in general, and the leading cause of disability, foot ulceration and ultimately amputation. Management of this disabling condition lacks either efficient or permanent results and is associated with significant costs. The major sign and symptom of painful DSPN is symmetrical sensory pain affecting the lower limbs, especially at nighttime. Assessment of DSPN is not difficult but making the correct diagnosis and choosing the right treatment are more challenging. The treatment of DSPN has three primary objectives: glycemic control, pathogenic mechanisms and pain management. Most often, guidelines recommend a symptomatic therapy aimed at pain reduction, with the use of antidepressants, anticonvulsants, opioids, topical treatment with capsaicin or lidocaine. Pathogenesis-oriented treatment is represented mainly by alpha-lipoic acid and group B vitamins supplementation.

Frequency Rhythmic Electrical Modulated System (FREMS) is a method of transcutaneous treatment based on frequency-modulated electromagnetic neural stimulation. It is proposed as an innovative and valid method of treatment of various disorders beginning from sports injuries (muscle injuries, tendinopathies, etc.), through musculoskeletal disorders, such as radiculopathies (cervicobrachialgia, low back pain, carpal tunnel, etc.), to peripheral neurovascular complications, such as diabetic neuropathy, vasculopathies (arteriopathies, venous or mixed), chronic ulcers of various etiology (traumatic, neuroischemic, arterial, venous, mixed, pressure or neuropathic ulcers). FREMS efficacy in neuropathic pain in diabetes mellitus still lacks enough research.

The primary endpoint of the study was the treatment’s effect on pain perception on the visual analogue scale (VAS) assessed each day in the morning. Secondary endpoints included changes in tactile sensation, temperature sensation and vibration sensation threshold on feet and changes in knee and ankle reflexes. Moreover, during the study, the patients’ quality of life was assessed by a five-level version of the EQ-5D (EQ-5D-5L). A seven-point Clinical Global Impression-Improvement scale (CGI-I) was used to track patients’ progress and treatment response over time.

## 2. Materials and Methods

Patients meeting the following criteria were enrolled to the study: (1) aged 18 years old and older; (2) diagnosed with diabetes mellitus type 1 or 2; (3) symptomatic diabetic polyneuropathy affecting the lower extremities with at least one positive sensory symptom, such as pain, burning, paraesthesia or prickling; (4) without changes in neuropathy treatment during last 30 days (stable doses of pain medications or other medications prescribed for diabetic neuropathy); (5) assessed by the investigator as able to maintain compliance and cooperation for 8 weeks of the trial. Exclusion criteria were as follows: (1) lack of informed written consent to participate in the study; (2) ankle-brachial index (ABI) < 0.5 suggesting ischaemia; (3) presence of an implanted pacemaker, defibrillator or neurostimulator; (4) treatment with TENS or other electrotherapy for diabetic neuropathy during last year; (5) pregnancy; (6) any concomitant severe disease limiting compliance to study procedures or life expectancy. The study participants were asked about the presence of neuropathy symptoms, such as burning, a painful feeling of cold or hot, pins and needles, electric shocks, pricking, numbness and itching. They underwent a neurological assessment as described below. The patients screened to the study were hospitalized in the Diabetology Clinic, Institute of Rural Health in Lublin, Poland for at least 5 days of treatment. The duration of the follow-up period was planned for the next 8 weeks.

The study was designed as a randomized, single-blind, sham-controlled, adjunctive therapy and single-center trial. After screening, eligible participants were randomized to the FREMS or sham-FREMS arm by drawing out of sealed white envelopes. The intervention itself lasted for 5 days and consisted of intravenous infusions of alpha-lipoic acid (600 mg q.d.) supplemented by active real FREMS therapy or an inactive procedure to mimic the active one. All the patients randomized were informed not to use any additional painkillers apart from acetaminophen while participating in the study. The patients received ten sessions of FREMS or sham-FREMS treatment spread over 5 days, each session twice daily, in accordance with the manufacturer’s recommendations for symptomatic diabetic peripheral neuropathy (DSPN). All patients underwent treatment in an isolated, quiet room, in a lying position. An Aptiva 4 device [Lorenz Lifetech, Ozzano dell’Emilia, Italy] with dedicated cables and four pairs of disposable electrodes applied to both lower extremities, were used, the screen facing away from the patient. The application of electrodes was on both legs and feet in accordance with the manufacturer’s instruction: on the tibial muscles (positioning red electrodes on the muscles and black electrodes on the tendon-muscle insertions), on the lateral surfaces of the calves (red electrodes proximally and black distally), below each malleolus (placing red electrode on the internal side of ankles) and on feet (placing red electrodes on the upper sides and the black ones on soles). FREMS used sequences of biphasic (negative and positive), asymmetric and electrically balanced pulses. One session comprised of two phases: an active and a recharging one. An active phase comprised of high negative voltage spike (variable, max −300 V) and extra short duration (variable, 10–100 μs, mostly ~40 μs). It was followed by a recharging phase of low voltage and long duration (0.9–999 ms) pulses of variable frequency ranging 1 to 1000 Hz, mainly in the low range 1–50 Hz. Every session duration was 35 min, including the first (active) phase lasting 25 min, then the second (recharging) phase lasting 10 min. In the sham-FREMS group, participants were connected to the running apparatus with the same type of dedicated electrodes, but without energy being released. The next part of the study was a follow-up lasting 8 weeks. Every participant was given a pain assessment diary to be filled out during the 5 days of treatment and another diary for the 8 week follow-up period. The patients assessed the intensity of pain in the VAS score every day in the morning throughout the study. VAS is one of rating scales used in epidemiologic and clinical research to measure the intensity of various symptoms, e.g., pain. The pain VAS is used widely to record patients’ pain intensity, progression or to compare pain severity changes. In our research we used the simplest 0–10 VAS as a straight horizontal line with numbers from 0 (no pain) to 10 (the worst pain in a lifetime). The patients also filled out a quality-of-life questionnaire (EQ-5D-5L) thrice: at the beginning of the study, after 5 days of therapy, after 8 weeks of follow-up and a Clinical Global Improvement scale (CGI-I) twice: after 5 days of therapy and after 8 weeks of follow-up. Every patient underwent an instrumental assessment of neuropathy twice, i.e., before and after 5 days of treatment. The technician who performed the instrumental neurological assessment was still the same, she did not administer the FREMS or placebo treatment and was unaware of the treatment assignment. Moreover, to secure the blindness of the study, participants were instructed that any perceptions or their absence during sessions are possible independently of the type of treatment given. The study protocol was approved by the Ethics Committee of the Institute of Rural Health and written informed consent was obtained from all participants before enrollment. The duration of the study was planned for 8 weeks to reduce the drop-out rate and to minimize the risk and need of additional pain medications intake. According to the literature available in this field it was the minimal time to obtain significant results.

Clinical data was acquired from hospital records; subjective information about neuropathy symptoms and their changes were collected from pain evaluation diaries and questionnaires filled out by patients; neuropathy assessment results were sourced from dedicated TSS (Total Syndrome Score) forms.

The treatment’s effect on pain perception on the visual analogue scale (VAS) was assessed each day in the morning. Results from the day after the 5-day treatment period and from the last day of 8 weeks of the follow-up phase were compared with baseline. Changes in tactile sensation were assessed by the 5.07 (10 g) Semmes–Weinstein monofilament test, changes in temperature sensation were assessed by the 10-point Tip-Therm test (warm–cold), changes in vibration sensation threshold on the participants’ feet were assessed using a manual biothesiometer. Knee and ankle reflexes were checked with a neurological hammer. The instrumental assessment of neuropathy and reflexes was conducted by the same person experienced in this field for years. A ten sensitive point test protocol was used for tactile and temperature sensation, with assessment at the big toe, third and fifth toe and three (first, third and fifth) metatarsal head points on both feet, followed by two points on metatarsus and one on the heel with three examinations per point, including one random false stimulation. Foot vibration perception threshold was assessed in two points per foot: a toe’s top and the lateral ankle. The effect of treatment was assessed separately for the tactile, temperature, vibration sensation and reflexes. For each participant the authors summed up the total scores in tactile sensation, temperature sensation, vibration perception and reflexes separately before and after five days of treatment. For tactile, temperature sensation and reflexes there was one point given when a feeling or reflex was present in certain location and zero points when it was absent. In tactile and temperature sensation the maximum score was 20 (10 points on each foot). For reflexes the maximum score was 4 (2 points on each leg). The vibration perception was assessed using a manual biothesiometer and there was an average score calculated from 4 examined locations. The authors compared the total or average scores achieved after five days of treatment with the output values for each participant and flagged them as “improvement” when the total number of points for a certain feeling or reflex was higher after treatment than before. Then, the Chi-squared test was used to compare the FREMS group with the sham-FREMS group.

Changes in quality of life were assessed by a five-level version of the EQ-5D (EQ-5D-5L) characterized by improved psychometric properties and validated in patients with diabetes [3]. This multi-attribute instrument considers five dimensions including mobility, self-care, usual activities, pain/discomfort and anxiety/depression followed by a vertical visual analogue scale used as a quantitative measure of health outcome reflecting the patient’s own judgment. Participants underwent the assessment three times, i.e., at the beginning of the study, after 5 days of treatment and after 8 weeks of the follow-up at the end of the study. The results from these three time points were compared between groups and between study phases. A 7-point Clinical Global Impression-Improvement scale (CGI-I) was used for the patient’s assessment of the improvement or deterioration degree and rated as: very much improved, much improved, minimally improved, no change, minimally worse, much worse and very much worse. The evaluation was done twice: after 5 days of therapy and after 8 weeks of follow-up.

Statistical analysis was conducted using Statistica v.13.0 (StatSoft) and GraphPad Prism v.9.0. MS Excel 2010 (Microsoft) was used to collect data and support statistical analyzes. In order to present the results obtained in the nominal and ordinal scale, the methods of descriptive statistics were used, i.e., the number (n) and the percentage (%). The results obtained in a quantitative scale were presented using the methods of descriptive statistics, i.e., arithmetic mean (M), median (Me), standard deviation (SD), quartile range (IQR), minimum (Min) and maximum (Max). To assess the compliance of the distribution of the examined variables with the normal distribution, the Shapiro–Wilk test was used. In the absence of a normal distribution of variables, non-parametric tests were used; while demonstrating the normal distribution of variables, parametric tests were used. The χ2 independence test was used to assess the relationship between the studied variables on the nominal and ordinal scale. A Wilcoxon pairwise test was used to evaluate the difference between the two measurements. The Student’s *t*-test for independent samples and the Mann–Whitney test were used to assess the difference between the two groups. The value of 0.05 (α = 0.05) was adopted as the critical significance level (α) for all tests. Based on the results of the analysis, the rules of test probability were adopted: *p* < 0.05—statistical significance, *p* < 0.01—strong statistical significance and *p* < 0.001—very strong statistical significance.

## 3. Results

Out of 45 patients screened in the study, 44 were randomized (one patient excluded due to the ankle-brachial index < 0.5). All of them had alfa-lipoic acid (600 mg q.d.) intravenous infusions given (standard of care, SOC). Twenty-four patients were allocated to the FREMS group (54.5%) with additional FREMS treatment and twenty to the sham-FREMS group (45.5%) with SOC only. All participants completed the 5 days treatment phase, filled out the diaries and questionnaires provided. In the 8 weeks follow-up phase, there were 23 patients assessed in the FREMS arm and 16 in the sham-FREMS arm as they completed all necessary forms (Table 1).

Baseline characteristics did not significantly differ between the two treatment groups (Table 2). The majority of patients in both groups were female, aged over 60 years old on average. The mean glycated hemoglobin level in the FREMS group was 8.5% (median 8.2%, minimum 6% and maximum 12.8%) vs. 8.5% in the placebo group (median 8.45%, minimum 6% and maximum 13.7%).

Considering the study results, in both treatment arms (FREMS + SOC vs. sham-FREMS + SOC) a statistically significant reduction in the level of pain perception (assessed with VAS) was found on the last day of hospital stay compared to day zero (*p* < 0.001) (Figure 1 and Figure 2). In the FREMS + SOC group, there was a 2.8 points improvement on the VAS scale, and 2 points in the sham-FREMS + SOC group.

However, after 5 days of treatment, there was no statistically significant difference (*p* > 0.05) in VAS changes between the FREMS and the sham-FREMS group. The reduction in the intensity of symptoms was obtained after average 3.9 days of FREMS and after 3.2 days in the sham-FREMS group. There was also no statistically significant difference in the number of days after which a reduction in the level of pain perception was observed (*p* > 0.05). Nevertheless, the statistically significant effectiveness of FREMS therapy versus sham-FREMS was observed on the last day of follow-up (*p* < 0.05), i.e., after 8 weeks (Figure 3). Figure 4 shows how the treatment arms diverge to reach statistical significance in the 8 weeks.

Moreover, the symptoms’ improvement assessed by the Clinical Global Improvement (CGI-I) rating scale was compared in two time points during the study. Despite the fact there were no statistically significant differences in the scores after the hospital treatment between the FREMS group and the sham-FREMS group (*p* > 0.05), the difference was noticeable after 8 weeks following the treatment period (*p* < 0.05).

Based on these two groups of premises, we conclude that the effectiveness of the add-on FREMS therapy measured by VAS and CGI-I lasted longer than alpha lipoic acid only (SOC) and was significant 8 weeks after its completion.

The participants judged the quality of life three times during the study. The evaluation done using the EQ-5D-5L questionnaire covered the following modalities: mobility, self-care, usual activities, pain/discomfort and anxiety/depression followed by a vertical visual analogue scale (VAS) used as a quantitative measure of general health outcome reflecting the patient’s own judgment. There were not any statistically significant differences in individual variables neither between groups nor the study time points in the treatment arms. However, the study showed general health improvement in the FREMS group based on changes in the vertical visual analogue scale being a part of EQ-5D-5L questionnaires. More specifically, after 5 days of neural stimulation, a statistically significant increase in the general health outcome (assessed by vertical VAS being a part of the EQ-5D-5L questionnaire, https://aci.health.nsw.gov.au/__data/assets/pdf_file/0003/632847/EuroQol-5-Dimension.pdf (Appendix A)) was demonstrated compared to the time before the treatments (*p* < 0.01). Still, significant results in vertical VAS persisted after 8 weeks of self-observation in FREMS group compared to the time before the procedures (*p* < 0.01) and compared to the time after 5 weeks of treatments (*p* < 0.05). In sham-FREMS group there were not any significant general health improvement assessed with vertical visual analogue scale (*p* > 0.05) in the time points described above.

As a secondary endpoint, the authors paid attention to the treatment influence on tactile, temperature and vibration sensation threshold on the participants’ feet. The neurological assessment before and after 5 days of treatment did not differ significantly neither within groups nor between them.

No treatment-related severe or non-severe adverse events were recorded during the study, either in the FREMS or in the placebo group.

## 4. Discussion

FREMS, as a method of neuropathy treatment, was based on TENS (transcutaneous electrical nerve stimulation) methodology. TENS involves the use of low-voltage electric currents to treat pain. The electrical stimuli in FREMS significantly differ from those commonly used in other known electrotherapies, such as TENS. FREMS provides sequences of biphasic electrical stimuli that vary simultaneously and automatically in frequency, pulse duration and amplitude, reaching relatively high intensity (300 V) with a very short duration (10–100 μs). It is through this method that it maintains electrical balance in treated tissues [4]. This feature is novel with respect to existing electrotherapies, which are normally based on ‘geometrical’ waveforms characterized by lower peak intensity and higher pulse duration. Additionally, specific new mechanisms of FREMS action have been described, such as microvascular blood flow enhancement [5], an increase in vasomotor activity mediated by smooth cells [6] and a release of vascular endothelial growth factors [7]. The studies mentioned collectively suggest a possible improvement in endoneurial blood flow induced by FREMS to explain its effects on neuropathy.

This research investigated the safety and efficacy of FREMS in a randomized, single-blind, sham-controlled, adjunctive therapy and single center clinical trial. FREMS was safe and resulted in a pain reduction in patients with symptomatic diabetic polyneuropathy. The effect was significant 8 weeks after the 5 days of treatment compared with SOC only. The FREMS treatment improved the participants’ quality of life.

The effect of FREMS on symptoms and signs of diabetic polyneuropathy was assessed in previous trials. However, their design and methods differ from those used in our research, which makes it difficult to compare the results. Below we summarize the most important ideas and results from the trials available so far, we hope that our paper could support the research on FREMS therapy in diabetic neuropathy.

In 2005, Bosi et al. from Italy published the study of the FREMS effectiveness in the treatment of painful diabetic neuropathy in a randomized, double-blind, crossover study with 31 patients involved [8]. Compared with our research, the treatment in Bosi’s study was twice as long as those applied in our study, i.e., each patient received two series of ten treatments of either FREMS or placebo. FREMS induced a significant reduction in daytime and night-time pain score (VAS) and a significant increase in sensory tactile perception (assessed by monofilament), a decrease in foot vibration perception threshold (measured by a biothesiometer) and an increase in motor nerve conduction velocity. The significant results persisted for 4 months of follow-up.

In 2013, Bosi et al. published another double-blind, randomized, multicenter study of even more, i.e., three series, each of ten treatment sessions of FREMS or placebo administered within 3 weeks, 3 months apart, with an overall, relatively long follow-up of about 51 weeks [9]. The primary endpoint included the change in nerve conduction velocity (NCV) of deep peroneal, tibial and sural nerves and the secondary endpoints were the treatment effect on pain, tactile, thermal and vibration sensations. In a total of 110 participants, FREMS had no effect on NCV but it induced a significant reduction in pain (VAS) after each treatment session, although this beneficial effect was no longer measurable three months after treatment. In the FREMS group the cold sensation threshold was significantly improved, while non-significant differences were observed in the vibration and warm sensation thresholds.

The two studies carried out by Bosi et al. [8,9] were partially consistent showing the need for further research. Both studies resulted in decreasing pain and improving peripheral sensation, whereas NCV results were different. Increasing the number of treatments in the second study did not prolong its effect. In our study, one series of ten treatments resulted in pain reduction and quality of life improvement observed after 2 months (8 weeks) from the treatment.

In 2021, Crasto et al. conducted a pilot, randomized controlled trial in the United Kingdom in patients with dual neuropathic pain treatments recruited from primary and secondary care [10]. Participants were randomized to FREMS and usual care (n = 13) or usual care only (n = 12). Primary outcome was a change from baseline in perceived pain (assessed by VAS) at 12 weeks between treatment groups. Similar to our results, FREMS group showed improvements in perceived pain compared with baseline, although the change in VAS was not statistically significant from the control group. Nevertheless, there were significant improvements in pain after FREMS therapy, assessed by McGill Pain questionnaire and Douleur neuropathique-4 questionnaire. More participants on FREMS had greater than 30% reductions in pain compared with the control group. Significant improvements in Patient Global Impression of Change were also noticed.

Health-related quality of life (HRQoL) has become an important health outcome indicator applied in many clinical trials [11] and focuses on factors that are part of the person’s health (well-being and functioning). One of the instruments developed to measure HRQoL is the EQ-5D-5L scale. It is the new version of EQ-5D scale used world-wide and may be used in the clinical care of patients with diabetes mellitus, also in Poland [1].

In our research, the applied therapies (FREMS, sham-FREMS, both with alpha-lipoic acid) did not show a significant improvement in mobility, self-care, usual activities nor a decrease in pain/discomfort, anxiety/depression, but FREMS therapy improved the global quality of life assessed immediately after therapy. This effect was maintained after 8 weeks in contrast to the standard of care therapy (sham-FREMS + SOC).

A strength of our study was its design, which included patients randomization, single-blind masking and a placebo (sham-FREMS) arm. A limitation of the study is a quite short follow-up period. The studies assessing FREMS therapy results should last longer to improve the outcomes. Moreover, some of the participants dropped out during the study. However, although the dropout rate during the study was not negligible, it was not treatment related, as suggested by the fact that it was higher in the placebo arm than in the FREMS arm (20% and 4%, respectively) and caused by the burden associated with study participation. What is more, our research lacked a neurological assessment after 8 weeks of follow-up. The results of the study also lack a gender distinction which may be important as DSPN is different between genders. Additionally, despite the fact that DSPN is strongly associated with glycemic control, the authors did not perform neither a blood glucose nor HbA1c tracking over the duration of the study. The research did not consider any other clinical data, such as dietary patterns, or types of anti-diabetic medications. The pain medication control was based only on the patients’ commitment of not using any additional pain medications apart from acetaminophen while participating in the study.

In our study, FREMS treatments, as an add-on therapy to alpha-lipoic acid intravenous infusions, resulted in decreasing pain severity, improving patients’ quality of life and clinical global impressions. This method of therapy needs further randomized controlled studies to confirm and strengthen this and previous reports. Additionally, the cost-effectiveness studies of different treatments for diabetic neuropathy are still needed [12].

## 5. Conclusions

In conclusion, the findings of the study confirm that FREMS is a safe method of alleviating pain in diabetic patients with DSPN and improves the patients’ quality of life. Currently, in the face of a still limited number of effective solutions it may turn out to be another building block in causal and symptomatic DSPN therapy. For patients who suffer from DSPN and have to deal with chronic peripheral pain, FREMS therapy may be an important complementary or alternative method of alleviating symptoms.

## Figures and Tables

**Figure 1 ijerph-20-00111-f001:**
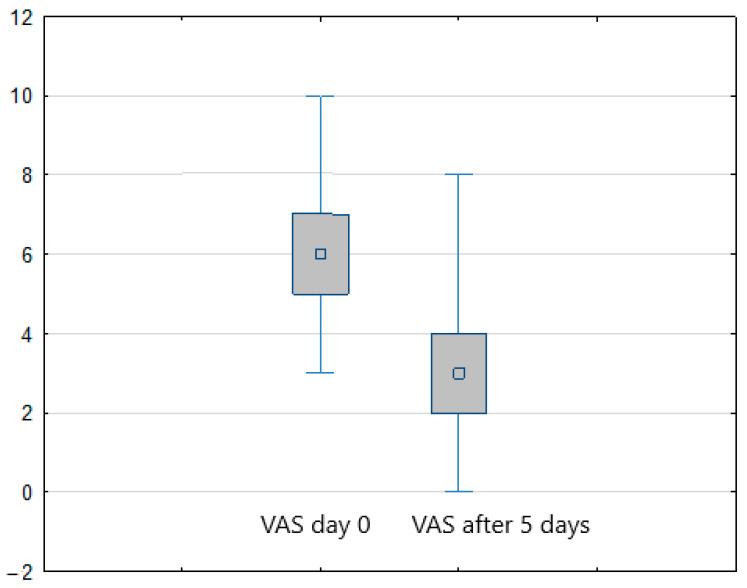
Reduction of pain (VAS score) in the FREMS (+ SOC) group after 5 days of treatment.

**Figure 2 ijerph-20-00111-f002:**
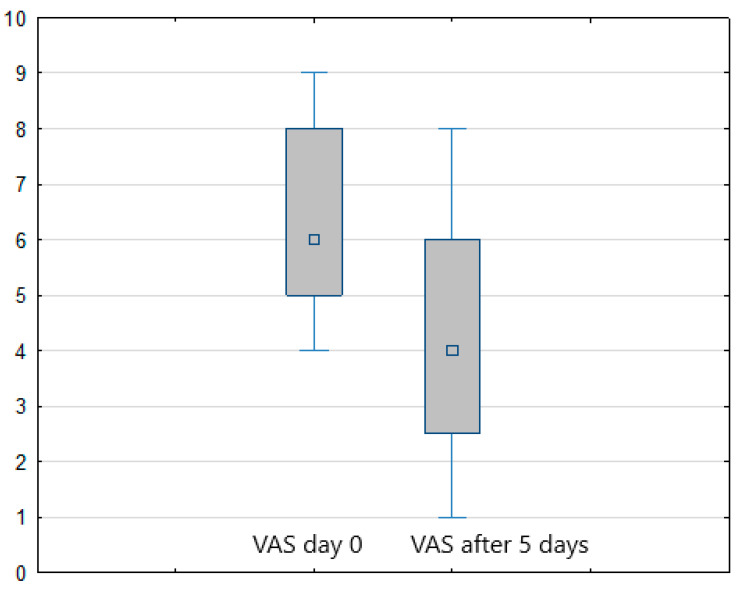
Reduction of pain (VAS score) in the sham-FREMS (+SOC) group after 5 days of treatment.

**Figure 3 ijerph-20-00111-f003:**
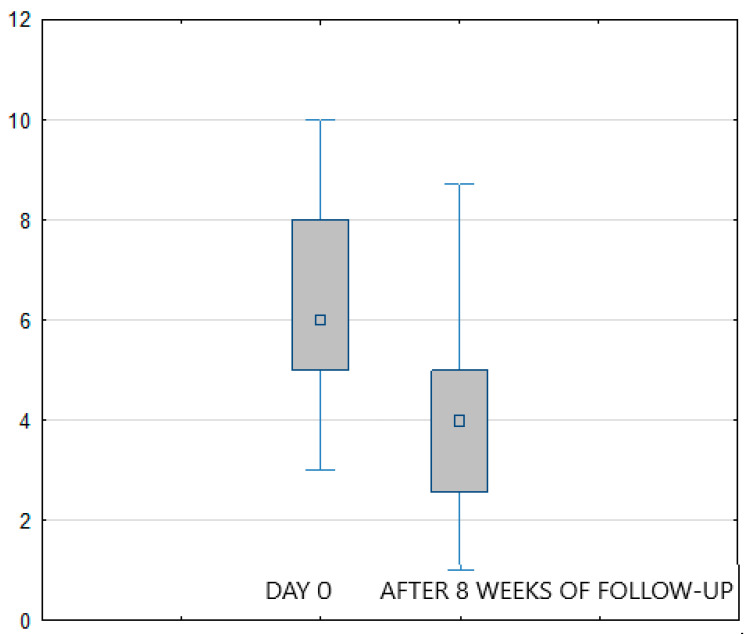
Reduction of pain (VAS score) in the FREMS group after 8 weeks of follow-up.

**Figure 4 ijerph-20-00111-f004:**
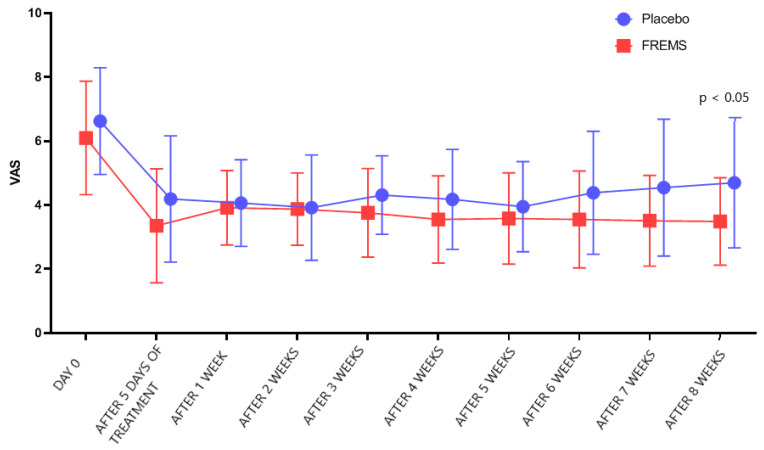
The trend of changes in VAS during the study on sham-FREMS and FREMS groups.

**Table 1 ijerph-20-00111-t001:** Flow diagram of the study.

**SCREENED:**	45 Patients	
**RANDOMIZED:**	44 patients	
**TREATMENT:**	FREMS	sham-FREMS
	24 patients (54.5%)	20 patients (45.5%)
**FOLLOW-UP:**	23 patients (96%)	16 patients (80%)

**Table 2 ijerph-20-00111-t002:** Baseline characteristics of the study participants.

	FREMS Group	Sham-FREMS Group
age [years]		
mean ± SD	64 ± 10.5	62 ± 11.5
median	66	65
min; max	37; 84	46; 80
duration of DM treatment [years]		
mean ± SD	17 ± 11.9	23 ± 10.7
median	16.5	22
min; max	1; 58	6; 49
HbA1c [%]		
mean ± SD	8.5	8.5
median	8.2	8.45
min; max	6; 12.8	6; 13.7
gender		
female	13 (54%)	12 (60%)
male	11 (46%)	8 (40%)

SD—standard deviation.

## Data Availability

Data supporting reported results available and owned by the authors.

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
