# Peer review of "The 8-Week Efficacy of Frequency Rhythmic Electrical Modulated System (FREMS) as an Add-on Therapy in the Treatment of Symptomatic Diabetic Peripheral Polyneuropathy"

_ijerph, 2022, doi:10.3390/ijerph20010111_

Round 1

Reviewer 1 Report

Gorczyca-Siudak_FREMS_IJERPH_2022

I commend the authors on the completion of this manuscript but it needs some modifications that I will point out in the comments sections below.

Specific Comments

Abstract: 

Please explain the next acronyms in the abstract: VAS, FREMS, EQ5D5L and CGI-C

Introduction

The introduction is too short. There are relevant references explaining the treatment basis for FREMS that should be included. For example: Bocchi L, Evangelisti A, Barrella M, Scatizzi L, Bevilacqua M. Recovery of 0.1Hz microvascular skin blood flow in dysautonomic diabetic (type 2) neuropathy by using Frequency Rhythmic Electrical Modulation System (FREMS). Med Eng Phys. 2010 May;32(4):407-13. doi: 10.1016/j.medengphy.2010.02.004. Epub 2010 Mar 6. PMID: 20207576.

I recommend also to add the background of the treatment in other clinical situations related to neuropathy and to specify the aim more clearly. 

Material and Methods

Line 56: Where was the placement of the electrodes made? 

Line 89 and next. In what way the total scores of the neuropathy assessment results were classified in “improvement” or “lack of improvement”? A statistical analysis with Wilcoxon test was made? And then the comparison of categories was assessed with the Chi-squared test? Please clarify here and in the statistical analysis section.  If not, the results presented in the line 188 are not clear. 

Results

Line 124. One person was not randomized, why? 

Line 153. Please change to Figure 5. 

Line 160. Legend. Please change to Figure 5. 

Line 163. The symptoms’ improvement assessed by the Clinical Global Improvement (CGI-I) rating scale was compared in two time points, not three, during the study. Please correct. 

Line 170. Please move this paragraph about intragroup changes of VAS near to the figure that represent the changes. 

Discussion

Lines 194 and 211. The two first paragraphs in the discussion section can be moved to the introduction section. 

Line 240. Please syntax review of this sentence. 

Lines 231 and 240. When you present the results from the studies of Bosi, because of the writing, it is not possible to know to what group refers the changed exposed. Are the changes you refer to intragroup of intergroup? Please clarify. 

Line 261. The changes you refer to after FREMS therapy, are after 12 weeks also? Are they intragroup or intergroup? Please clarify. 

Line 263. Changes, intra or intergroup? 

Line 273. But without intergroup differences? Please clarify. Here and also in the results and conclusion section. It seems like you have mixture in the conclusion section the results which are related to the time line (intragroup), with the differences between groups (intergroup). 

Conclusion

Line 289. Please change the last sentences of the conclusions to the end of the discussion section. 

References

Please, change the format of the references to the standards of the Journal. 

Author Response

Dear Reviewer,

First, we would like to thank you very much for the helpful comments. We’ve changed the paper as suggested. Also, some explanations are given below in italics.

1. Please explain the next acronyms in the abstract: VAS, FREMS, EQ5D5L and CGI-C - Done

2. Introduction has been changed and extended.

3. Material and Methods

Line 56: Where was the placement of the electrodes made? - Explained

Line 89 and next. In what way the total scores of the neuropathy assessment results were classified in “improvement” or “lack of improvement”? A statistical analysis with Wilcoxon test was made? And then the comparison of categories was assessed with the Chi-squared test? - Done

4. Results

Line 124. One person was not randomized, why? - Explained

Line 153. Please change to Figure 5. - Done

Line 160. Legend. Please change to Figure 5. - Done

Line 163. The symptoms’ improvement assessed by the Clinical Global Improvement (CGI-I) rating scale was compared in two time points, not three, during the study. Please correct. -Done

Line 170. Please move this paragraph about intragroup changes of VAS near to the figure that represent the changes. - Changed

5. Discussion

Lines 194 and 211. The two first paragraphs in the discussion section can be moved to the introduction section. - Changed

Line 240. Please syntax review of this sentence.  - Changed

Lines 231 and 240. When you present the results from the studies of Bosi, because of the writing, it is not possible to know to what group refers the changed exposed. Are the changes you refer to intragroup of intergroup? Please clarify.  Line 261. The changes you refer to after FREMS therapy, are after 12 weeks also? Are they intragroup or intergroup? Please clarify.  Line 263. Changes, intra or intergroup? Line 273. But without intergroup differences? Please clarify. Here and also in the results and conclusion section. It seems like you have mixture in the conclusion section the results which are related to the time line (intragroup), with the differences between groups (intergroup). 

 --- > The design and methods of the papers in discussion differ from those used in our research, which makes it difficult to compare the results. We’ve summarized the most important ideas and results from the trials. We’ve added lines 291-294 to explain it.

6. Conclusion

Line 289. Please change the last sentences of the conclusions to the end of the discussion section.  - Done

7. References

Please, change the format of the references to the standards of the Journal. - The references were formatted (and now changed) according to the sites: https://www.mdpi.com/journal/ijerph/instructions, https://www.mdpi.com/authors/references and as in the example:

Journal Articles:
1. Author 1, A.B.; Author 2, C.D. Title of the article. Abbreviated Journal Name YearVolume, page range.

Reviewer 2 Report

The reviewer would suggest to consolidate the introduction of the manuscript that does not provide the research literature context. In addition the conclusion should not be a state of the art but more an in depth discussion of the results to explain them.

Nonetheless the issue with the present work is the too short duration of the assay to validate the result obtained by the end of the study. Due to this issue the assay appears as more a negative result than a positive one even though data suggest the potential of the procedure.

Author Response

Dear Reviewer,

First, we would like to thank you very much for the helpful comments. We’ve changed the manuscript as suggested by reviewers. The aim of the paper was to constitue a part of the research on FREMS therapy in diabetic neuropathy even if the results are not groundbreaking. In our opinion, giving even more a negative result than a positive one could also be helpful in placing the role of FREMS in DSPN treatment.

Reviewer 3 Report

The article under adjudication, reports FREMS treatment on DSPN. Although the article has merit, however, there are some major limitations that needs to be addressed for further review process.

1) Kindly improve the entire structure and organization of the manuscript in the form of a standard publishable article. The Introduction needs to be refurnished with materials from discussion/conclusion section and literature review. DSPN literature references must be updated. Currently ~40-60% patients with T2DM are diagnosed with DSPN - I disagree with line 200.

2) In the methods, kindly add a flow diagram mentioning how subjects were recruited and screening criteria if any. FREMS method, dosage of alpha lipoic acid, assessment of DSPN (to begin with), exact questionnaires of the pain assessment diary (based on which the VAS scores were calculated), any available TSS scores must be disclosed as part of the manuscript and evaluations. The terms VAS must be clearly defined for a general readership, as it is the only endpoint used to estimate a difference.

3) DSPN progression is starkly different between genders. Is it possible for the authors to categorize their results based on that?

4) Fig 1 is missing in manuscript. From Fig 2 and 3, it is quite evident that alpha lipoic acid is causing a much more severe change in VAS as compared to FREMS. I am quite intrigued by this and was hoping if the authors may enlighten. In addition to Fig 4, I am missing a reduction in VAS Score for the Sham group over 8 weeks follow up. 

5) DSPN and its control is strictly restricted to primary glycemic control of patients. Is it possible for the authors to provide such clinical data over 8 weeks span for the subjects?

6) Why was the study duration fixed for 8 weeks, when there is a clear trend that the outcome may have improved if the study lasted longer than 8 weeks time frame (Fig 4 reference)? I do acknowledge the dropout rate, but I do question the reason only 8 weeks were chosen.

7) Is there any other available clinical data relevant to the subjects available? For eg, dietary patterns, type of anti-diabetic medications and/or if any pain medications were taken during the course of the study etc. Kindly furnish the manuscript to strengthen the outcomes. 

8) Although the authors may be experts in the field of diabetic neuropathy, I am requesting them to look for references from https://clinicaltrials.gov/ct2/show/NCT01628627 to help develop this article.

Author Response

Dear Reviewer,

First, we would like to thank you very much for the helpful comments. We’ve changed the manuscript as suggested by reviewers. Also, some explanations are given below in italics.

1) Kindly improve the entire structure and organization of the manuscript in the form of a standard publishable article. The Introduction needs to be refurnished with materials from discussion/conclusion section and literature review. DSPN literature references must be updated. Currently ~40-60% patients with T2DM are diagnosed with DSPN - I disagree with line 200. – It has been changed.

2) In the methods, kindly add a flow diagram mentioning how subjects were recruited and screening criteria if any. - The flow diagram is present in the results.

FREMS method, dosage of alpha lipoic acid, assessment of DSPN (to begin with), exact questionnaires of the pain assessment diary (based on which the VAS scores were calculated), any available TSS scores must be disclosed as part of the manuscript and evaluations. The terms VAS must be clearly defined for a general readership, as it is the only endpoint used to estimate a difference. - Improved.

3) DSPN progression is starkly different between genders. Is it possible for the authors to categorize their results based on that? - We didn’t perform such statistics for the purpose of this study. I could be included in furhter research with a larger group of participants.

4) Fig 1 is missing in manuscript. - It has been present – after corrections line 181

From Fig 2 and 3, it is quite evident that alpha lipoic acid is causing a much more severe change in VAS as compared to FREMS. I am quite intrigued by this and was hoping if the authors may enlighten. In addition to Fig 4, I am missing a reduction in VAS Score for the Sham group over 8 weeks follow up. - We’ve changed this section to be more comprehensive and understandable.

5) DSPN and its control is strictly restricted to primary glycemic control of patients. Is it possible for the authors to provide such clinical data over 8 weeks span for the subjects? We didn’t perform such statistics for the purpose of this study. It could be included in furhter research.

6) Why was the study duration fixed for 8 weeks, when there is a clear trend that the outcome may have improved if the study lasted longer than 8 weeks time frame (Fig 4 reference)? I do acknowledge the dropout rate, but I do question the reason only 8 weeks were chosen. - The explanation has been added in line 116-118

7) Is there any other available clinical data relevant to the subjects available? For eg, dietary patterns, type of anti-diabetic medications and/or if any pain medications were taken during the course of the study etc. Kindly furnish the manuscript to strengthen the outcomes.  - Unfortunately, we didn’t perform such statistics for the purpose of this study. When it comes to pain medications it is explained in the paper – line 78-80

8) Although the authors may be experts in the field of diabetic neuropathy, I am requesting them to look for references from https://clinicaltrials.gov/ct2/show/NCT01628627 to help develop this article. - We’ve already used the results of that clinical trial of Bosi et al in the discussion and references.

Round 2

Reviewer 1 Report

Gorczyca-Siudak_FREMS_IJERPH_2022

I thank the authors the effort to implement almost all the corrections suggested. However, there are still some remained changes to afford.

Specific Comments

Introduction

Please clarify the aim at the end of the introduction section and not in the material and methods section. 

Material and Methods

Line 124 and next. The way the total scores of the neuropathy assessment results were classified in “improvement” or “lack of improvement” is not clear yet. Moreover to what variables you used Wilcoxon test? To what variables you used Chi-squared? Why you don’t integrate this statistical analysis with the other statistical analysis section and in this part of the material and methods explain clearly the way the variables were registered and categorized? 

Results

Line 203. Figure 3 legend: Part of the text of the legend must be in the main text after the figure. 

Line 234. Please clarify that these changes in VAS, only and not in the other subscales of the questionnaire, were between treatment arms. 

References

Please, change the format of the references to the standards of the Journal. 

Author Response

Thank you very much for your effort and the comments. We’ve tried to implement all corrections suggested by reviewers (red font in the text). Also, some explanations are given in italics as below.

Introduction

Please clarify the aim at the end of the introduction section and not in the material and methods section. - done, lines 55-61

Material and Methods

Line 124 and next. The way the total scores of the neuropathy assessment results were classified in “improvement” or “lack of improvement” is not clear yet. Moreover to what variables you used Wilcoxon test? To what variables you used Chi-squared? Why you don’t integrate this statistical analysis with the other statistical analysis section and in this part of the material and methods explain clearly the way the variables were registered and categorized? - lines 150-163 modified; we aimed to group statistical methods in one place in the text

Results

Line 203. Figure 3 legend: Part of the text of the legend must be in the main text after the figure. - corrections done

Line 234. Please clarify that these changes in VAS, only and not in the other subscales of the questionnaire, were between treatment arms. - clarified: lines 260-262

References

Please, change the format of the references to the standards of the Journal. - modified

Reviewer 2 Report

Thank you for this new version with the improved introduction.

The reviewer suggest to make an effort on the discussion as mentionned in the precedent message

Author Response

Thank you for all your comments. We’ve tried to implement all specific corrections suggested by all reviewers which have changed the manuscript as you can see in attached file.

Reviewer 3 Report

After thoroughly evaluating the revised manuscript and the author responses, I understood, that the manuscript cannot be further improved as the design of the study was not appropriate and/or I doubt if the study was initially considered to be for a publication. However, I do acknowledge text improvement and the relevant changes in the manuscript does substantiate the presentation of the work.  I thank the authors for consideration and response to the reviewer comments. Further, I would like to add a couple of minor points - 
1) The title of the manuscript at the moment is too redundant. Kindly re-format to a short title with mention of the 8 weeks duration of the study - this would be to avoid any mis-consideration the field of diabetic neuropathy.

2) All limitations of the article must be clearly mentioned in the discussion section. For eg, a gender based distinction, blood glucose tracking over the duration of the study or any other sensory tests were not performed. A lot of the short-comings were addressed by the authors as a 'future consideration' for my previous round of revision. Kindly incorporate them.

3) Line 324-328 clearly brings out the essence of the entire study but only mentioned at the very end of the manuscript. Kindly ensure this is mentioned in the study design, again to avoid any misconception.

4) Kindly provide quality-of-life questionnaire (EQ-5D-5L) as SI.

Thank you.

Author Response

Thank you very much for all your comments. We’ve tried to implement all corrections suggested by reviewers (red font in the text). Also, some explanations are given in italics as below.

1) The title of the manuscript at the moment is too redundant. Kindly re-format to a short title with mention of the 8 weeks duration of the study - this would be to avoid any mis-consideration the field of diabetic neuropathy. - modified

2) All limitations of the article must be clearly mentioned in the discussion section. For eg, a gender based distinction, blood glucose tracking over the duration of the study or any other sensory tests were not performed. A lot of the short-comings were addressed by the authors as a 'future consideration' for my previous round of revision. Kindly incorporate them. - modified, lines 350-358

3) Line 324-328 clearly brings out the essence of the entire study but only mentioned at the very end of the manuscript. Kindly ensure this is mentioned in the study design, again to avoid any misconception. - added, lines 80-81 and 129-130

4) Kindly provide quality-of-life questionnaire (EQ-5D-5L) as SI - provided
